# Hierarchical Planning for Rope Manipulation using Knot Theory and a Learned Inverse Model

**Matan Sudry, Tom Jurgenson, Aviv Tamar and Erez Karpas**
Technion — Israel Institute of Technology, Haifa, Israel
{matansudry, tomj}@campus.technion.ac.il, {avivt, karpase}@technion.ac.il

**Abstract:** This work considers planning the manipulation of deformable 1-dimensional objects such as ropes or cables, specifically to tie knots. We propose TWISTED: Tying With Inverse model and Search in Topological space Excluding Demos, a hierarchical planning approach which, at the high level, uses ideas from knot theory to plan a sequence of rope topological states, while at the low level uses a neural-network inverse model to move between the configurations in the high-level plan. To train the neural network, we propose a self-supervised approach, where we learn from random movements of the rope. To focus the random movements on interesting configurations, such as knots, we propose a non-uniform sampling method tailored for this domain. In a simulation, we show that our approach can plan significantly faster and more accurately than baselines. We also show that our plans are robust to parameter changes in the physical simulation, suggesting future applications via sim2real.

**Keywords:** Knot tying, Learning, Planning

## 1 Introduction

Deformable object manipulation is important for many applications, such as manufacturing and robotic surgery. In particular, manipulating 1-dimensional (1D) objects such as ropes, cables, and hoses, is a challenging and exciting research area that has drawn recent attention [1, 2, 3, 4, 5, 6, 7, 8, 9, 10]. There are several challenges to 1D object manipulation.

Representing the state of the object is difficult, as unlike rigid objects, the object may have infinite degrees of freedom [11, 12, 13]. Perception of a rope-like object is complex due to self-occlusions, the similarity between different rope parts, and self-loops [1, 14, 15, 16, 17, 18]. Planning typically requires an effective abstraction of the states and the actions, which may be difficult to define [19, 20], and low-level control for executing a plan must handle the flexibility and deformability of the rope – all non-trivial control problems [3, 21, 22]. To the best of our knowledge, a system that can generally manipulate 1D objects is beyond the capabilities of current technology. Our focus in this work is on the planning component in 1D manipulation, particularly rope manipulation and knot tying. As mentioned above, planning for rope manipulation is non-trivial, as the state space may be large or infinite, and tasks like knot-tying essentially have a 'needle in a haystack' characteristic and require exhaustive exploration to reach desired states. Accordingly, most prior studies on rope manipulation relied on *human demonstrations* in lieu of automatic search [3, 23, 5, 24, 25, 6, 26, 27].

This paper tackles the problem of rope manipulation planning without any demonstrations. Our main contribution is a hierarchical search algorithm that exploits prior knowledge about knot-tying geometry for its high-level plan, with self-supervised learning of an inverse model for executing the low-level control, which we call Tying With Inverse model and Search in Topological space Excluding Demos – TWISTED. TWISTED is trained and evaluated in a physical simulation. We demonstrate, however, that our planning results are robust to variations of physical properties such as friction. Thus, we believe that our planning approach can be integrated with real-robot perception and control for a complete 1D manipulation system in the future. We demonstrate that TWISTED

7th Conference on Robot Learning (CoRL 2023), Atlanta, USA.

can tie various types of knots and can generalize to tie knots that were not seen during the training. To the best of our knowledge, this is the first demonstration of such a capability, which cannot be obtained by previous work that required a human demonstration of the knot to tie. Finally, while TWISTED is tailored for knot tying by building on knot theory for high-level planning, our general methodology may be useful for other tasks where a well-established theory may inform the high-level characterization of the problem, while a learning-based method is used for low-level control.

## 2 Background

In our work, we build on knot-theory for high-level planning. In this section, we give a brief overview of knot theory. The most common way to solve problems like knot-tying, with a high-dimensional and continuous state and action spaces, and long-horizon planning is to separate it into a topological representation for high-level planning and a geometric representation for low-level control, which is solved using learning [24]. We represent a rope as having $L$ links, and denote by $q \in Q$ the rope configuration, with $Q \subseteq$

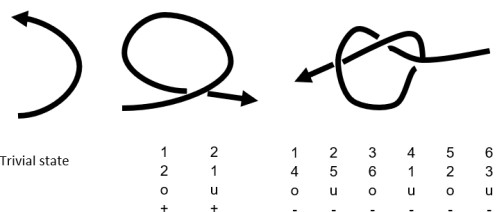

Figure 1: P-data Topological state representation: each column corresponds to an intersection along the rope of $L$ links. Row one is ordered from 1 to $L$ in ascending order. Row two, for each intersection, defines the other link in the intersection. The label "o"/"u" classifies the vertical arrangement at each intersection (over or under). Finally the last row identifies the "sign" – see appendix Section 9.1. E.g. in the center state representation link 1 is *over* link 2 with a "+" sign

$\mathbf{R}^{2L+5}$. The first seven coordinates describe the global position of the middle rope link (position $(x, y, z)$ and quaternion representation for the rotation), and the remaining $L - 1$ joints are each described by yaw and pitch values of the $i - 1$ link relative to link $i$.

We follow Yan et al. [3], where the discrete topological representation for $S$ is **P-data** (see appendix Section 9.1), and we denote Top:$Q \to S$ the mapping from a configuration to its topological state. The "complexity" of a topological state $s \in S$ is defined according to the number of crosses (link intersections, see Section 9.1) it represents figure 1. Knot theory [28] suggests *Reidemeister moves* as actions that transition the rope between topological states. In this work, we will use them as high-level actions during the search. We denote the space of Reidemeister moves as $A_R$ and $P_R :$ $S \times A_R \to S$ as the transition function of topological states using Reidemeister moves. Reidemeister [28] proved that between any two topological states $s, s' \in S$, there exists a sequence of actions that starts in $s$ and ends in $s'$, namely, $\exists a_0, \ldots, a_k \in A_R$ s.t. $S' = P_R(\ldots P_R(P_R(s, a_0), a_1) \ldots, a_k)$. The Reidemeister moves are (1) Reidemeister I (R1) which moves one segment to create a new loop, (2) Reidemeister II (R2) which pulls the middle segment and creates a new intersection with opposite signs, and (3)the Cross (C) creates a new intersection between two segments. Examples of those actions can be visualized in Figure 4 in the appendix.

Considering the knot-tying problem as a trajectory over topological states with Reidemeister moves as actions, translates the original problem of directly manipulating rope configurations to a problem of a shorter horizon and a "lower" branching factor. This approach has been adopted in different algorithms [3, 23, 5, 24, 25].

Finally, we use the **topological motion primitives** action space [3]: When manipulating a rope with $L$ links, an action is a curve $c \in C$, parameterized by the link to grab $l \in [1, L]$[1], an endpoint in $(x, y)$ (in the workspace), and the maximal height $z_{\max}$. We denote the transition function for curves applied on configurations by $f : Q \times C \to Q$. Yan et al. [3] observed that the space of curves $C$ approximates well all the possible Reidemeister moves available from a given topological state.

---

[1]We associate a fixed point for every link.

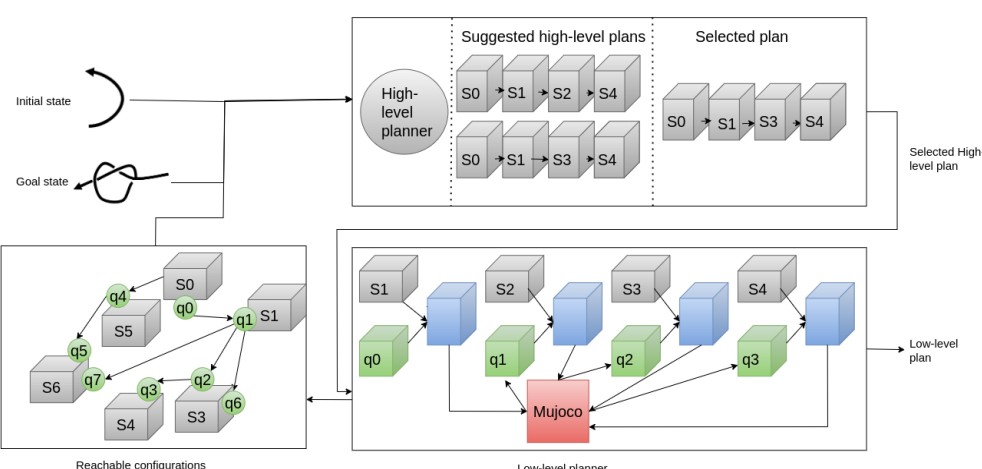

Figure 2: TWISTED: Given initial and goal topological states, we iteratively call a high-level planner to find plans to follow (top row). The plan uses an inverse model to transition between consecutive topological states (bottom right). When following a plan, new information is integrated into a tree of all known configurations, which seeds the high-level planner with initial states (bottom left). Gray boxes are high-level states, green boxes are low-level states (rope configuration), blue boxes represent the inverse model, and our environment in Mujoco is red. See Section 4.

For this reason, in this work, we follow Yan et al. [3], and plan using Reidemeister moves while manipulating the rope with curves.

# 3 Related work

Manipulating deformable objects presents varying degrees of difficulty [29]. For rope manipulation, recent studies focused on either learning from human demonstrations [2, 6, 3] or solving short-term plans (e.g., changing the shape of the rope, but not tying a knot) through pick-and-place actions [7, 27]. Differently, in our work we tackle long-term planning, such as knot tying, without demonstrations. To handle the challenging deformable-object dynamics of ropes, previous methods used self-supervised learning [10, 26, 17]. We also use self-supervised collected data to train our inverse model. Finally, some previous works attempted to learn rope manipulation using reinforcement learning (RL) methods [30, 31, 32]. However, as we show, our knot-tying tasks are too complicated for off-the-shelf RL algorithms (see Section 5).

Several recent works studied the simpler task of rope un-tying [33, 34, 35, 15, 36]. In particular, [33, 34] used graph-based search algorithms. Knot un-tying is simpler because the end goal is always the same (untangled rope), while in our work we specifically focus on a large space of possible goals, which renders un-tying methods inapplicable.

# 4 TWISTED

In this section we describe the components that comprise our solution – TWISTED. We start with the description of the simulated environment in Section 4.1, we follow with the description of the algorithmic components in Sections 4.2, 4.3, and 4.4, and finally describe our data collection methodology 4.5. See Figure 2 for an overview.

## 4.1 Simulated Environment

The environment we used to learn and test TWISTED was created by the free, open-source simulation environment of Mujoco [37]. It includes the default rope of Mujoco and the end-effector moving the rope. We used a free-moving end-effector to focus on the complexity of tying knots, ignoring the additional complexity of controlling a robot manipulator[2]. It is crucial to mention that

---

[2]Although non-trivial, we expect that common motion planning solutions could be utilized in order to bridge the gap from a free-moving end-effector to a complete robotic manipulator.

during planning, we run actions in the simulation itself, meaning that both evaluation and search use the same Mujoco environment (i.e., the search acts with a perfect world model).

## 4.2 Planning Algorithm

TWISTED[3] is best summarized as an iterative algorithm that given an initial rope configuration $q_{init}$ and goal state $s_g$ repeats three steps: (1) starts searching from a known *reachable* configuration, (2) plans a high level trajectory whose states are in $S$, and actions in $A_R$, and (3) uses a low-level planner to follow subsequent states in the selected high-level plan. The iterative process of TWISTED repeats until $s_g$ is reached (success) or a pre-specified timeout expires (terminating in failure). See Algorithm 1.

**Data structures**: we maintain two data structures, a tree of known reachable configurations and a set of high-level plans. The *known reachable configurations*, is a tree $T$ whose vertices are configurations of the rope with their corresponding topological states $(q, \text{Top}(q))$ and the edges are low-level actions in $C$. Initially, $T$ contains only a root node - the rope's initial configuration $q_{init}$ and its topological state $s_{init}$. We also maintain a list of *high level plans*, $\mathbf{P} = \{P_i = (s_0, s_1, \ldots s_{l_i} = s_g)\}_i$ from *currently reachable topological states* (see Section 4.3). When a topological state $s'$ is discovered for the first time by the low-level planner (Section 4.3), we run the high-level planner from $s'$ and store the results into $\mathbf{P}$.

---
**Algorithm 1** TWISTED algorithm

---
**Input** $q_{init}$ Initial configuration state and $s_g$ topological goal state
**Output** Low-level plan if found

1: $init : T, \mathbf{P}$         ▷ see **data structures**
2: $s_{init} = Top(q_{init})$
3: populate $\mathbf{P}$ with plans from $s_{init}$
4: **while** Not timeout **do**
5:     $s_{slct} = SelectTopologicalState()$
6:     $P_{slct} = SelectPlan(s_{slct})$
7:     $q_{slct} = SelectConfig(s_{slct})$
8:     $PlanFound = FollowPlan(q_{slct}, P_{slct})$
9:     **if** $PlanFound$ **then**
10:       $ReturnPlan$
11:     **else**
12:       $RandomExpand()$
13:     **end if**
14: **end while**

---

**Plan selection:** At the start of each iteration, we need to select a plan to execute from $\mathbf{P}$ and a configuration to start executing the plan from. One naive heuristic is to select a random configuration from the reachable configurations. However, due to the problem's sparsity, we observe that during the search, configurations in $T$ with more crosses are exponentially fewer than those with fewer crosses. We thus seek an approach that promotes configurations corresponding to topological states with higher crosses. We therefore run the following three sub-procedures in sequence:

$SelectTopologicalState()$: identifies the reachable topological states in $T$ that have a high-level plan to the goal. Samples one such topological state $s$ according to one of two heuristics; *random*, which is the uniform distribution[4], and *prioritize-crosses*, which assigns $s \in S$ probability that is proportional to $\text{Cross}(s)$ (i.e. prefers topological states with more crosses – motivating to search deeper than the *random* heuristic).

$SelectPlan(s)$: a high-level plan (sequence of topological states) $P = s, s_1, \ldots s_l = g$ is randomly selected from all the high-level plans that start in $s$.

$SelectConfiguration(s)$: randomly select a configuration from all configurations belonging to $s$.

Plan execution: Next, in $FollowPlan$, we follow the high level plan $P = s_0, s_1, \ldots s_l$, where $s_0 = s$ and $s_l = s_g$, starting in $s$, and incrementally try to reach $s_{i>0}$ until $s_g$ is reached. To transition from $s_i$ to $s_{i+1}$, we use the low-level planner (Section 4.3) that uses the learned inverse model (Section 4.4) to predict curves. The low-level planner applies multiple curves $\{c_j\}_j$ to the current configuration $q_i$. Let $q'_j = f(q_i, c_j)$, and $s'_j = \text{Top}(q'_j)$. If the transition for $c_j$ reaches a configuration with more crosses, we add this information to $T$, and note that it could be the case that

---
[3]https://github.com/matansudry/twisted
[4]Even the *random* heuristic prioritizes complex topological states, as sampling a random topological state induces a different distribution than sampling from all reachable configurations in $T$ directly.

$s'_j \neq s_{i+1}$ because the inverse model is not perfect. Nevertheless, this is an executable low-level action; thus, we add it to our search tree.

Completeness guarantee: Finally, for completeness of the algorithm, after every iteration of TWISTED, with probability $p$ (hyper-parameter, with a default value of 0.05), we sample a random reachable configuration $q \in T$, execute $k = 100$ random actions and add them to $T$ if the action transitions to a topological state with a greater or equal number of crosses (same conditions as in the "plan execution" above). This ensures that given enough time, our algorithm is guaranteed to find a solution. We denote this sub-routine as $RandomExpand$.

### 4.3 Planning and Search

TWISTED is composed of two levels of planning, high-level and low-level planning, that are called as sub-procedures by the algorithm. We now describe the two planners with their states and actions.

`High-level planner`: with a state space $S$ and Reidemeister moves $A_R$ as actions, finds all paths from currently reachable topological states $s \in T$ (not necessarily $s_{initial}$) to $s_g \in S$ using Breadth-first search (BFS). Our BFS prunes new states $s'$ with $\texttt{Cross}(s') > \texttt{Cross}(s_g)$, and returns a set (possibly empty) of all paths that start in $s$ and terminate in $s_g$.

`Low-level Planner`: Given $s$ and $s'$ two consecutive topological states in the high-level plan, we search for a curve $c \in C$ that traverses from $s$ to $s'$. To successfully *and efficiently* find such a curve, we utilize our inverse model (Section 4.4) and generate curves $\{c_i \in C\}^K$ ($K = 6$). If any of the newly found topological states are $s'$, we return success (if more than one action succeeds we use the first one found), and the plan execution will move to the next topological state in the high-level path. Otherwise, we return failure, and the iterative process of TWISTED repeats.

### 4.4 Inverse model

An action generator is crucial in knot-tying as the proportion of curves that transition the rope to a given topological state could be extremely small (See Section 5.1 where we show the data collection difficulties). This makes it unlikely that a small set of randomly selected curves could be found to satisfy the required transition between topological states. Thus, we trained an inverse model to generate action candidates that are likely to satisfy the required transition. The inverse model is an auto-regressive model [38] predicts elements as follows: link to pickup, the height of the curve $z_{\max}$, destination X position, and destination Y position. The link is a categorical and modeled as a multinomial distribution, and the other elements are continuous and modeled with a Normal distribution. Every element is predicted with an *independent* sub-network, whose inputs are: (1) the current configuration, (2) the current $(x, y, z)$ coordinates of each of the $L$ links, (3) the **next** topological state $s'$, and (4) all the elements before the current element (e.g. $z_{max}$ gets the link index as input). See Figure 5 in appendix section 9.4.

**Training:** we collected data generated from *random actions* to train the inverse model (see Section 4.5). The data $D$ contains transitions $(s, s', q, c)$, $s$ and $s'$ are current and following topological states, $q$ is the current configuration, and $c$ is the curve taken. Since the data collection is time-consuming, we follow previous work of Yan et al. [3] and apply the Mirror and Reverse augmentations to our data.[5] We train the model via a maximum likelihood objective on $D$ (predicting $c$).

**Inference:** during inference, we follow the standard ancestral sampling scheme for auto-regressive models [38]; we predict a distribution for every element, sample from it, and feed the result to predict the next element in the sequence.

### 4.5 Data collection

To train the inverse model, we must collect data that represents movements typically encountered when tying knots. The problem, however, is that without a controller that knows how to tie knots, nor human demonstrations, it is not clear how to collect such data. Initially, we tried to collect rollouts simply by executing random walks of curves. However, in doing so, we found a very low

---

[5]In Yan et al. [3] these augmentations were applied over manual demonstrations. In our work, we apply them on randomly collected data.

number of topological states with three crosses (only 27 states per CPU core *per hour*), demonstrating that applying random actions to the rope typically does not lead to knot-like configurations. We, therefore, designed a collection scheme that selectively resets the environment. Using our scheme, we collected 537 successful transitions per hour per CPU core. We used that to collect a data set of 1,670,000 data points. For full details see Appendix 9.3.

## 5 Experiments

The experiments aim to address the following items: (1) How sensitive is knot-tying planning to the action space, and is a continuous action space necessary? (2) Comparison of TWISTED with baselines (3) How sensitive is TWISTED to changes in the physical simulation? (4) How well does TWISTED generalize to unseen knots?

### 5.1 What makes knot-tying difficult?

One difficulty of our knot-tying problem is that it requires very accurate actions to solve. To demonstrate this, we verify that even a fine discretization of the problem leads to significantly different outcomes. In this experiment, we measure sensitivity to discritization of curves, i.e. test if using a discretized curve reaches the same topological state as the next topological state (obtained by executing the original non-discretized curve). Formally, given a curve, $c = (i, z_{\max}, x, y) \in C$, which includes three continuous elements ($z_{\max}$, $x$, and $y$), we convert it to a discrete curve where each element is rounded. $z_{\max}$, is discretized in steps of 0.001, and x and y in steps of 0.01. Notice that the size of the discretized curve space is $21 \times 70 \times 100 \times 100 = 14,700,000$, already rather large. We measure the accuracy of the resulting topological states. If the accuracy is high, there is little difference in discretizing the action space, suggesting that knot tying could be solved using off-the-shelf discrete planners [39, 40]. We ran over 600k data points of transitions from topological states of two crosses to topological states of three crosses, and got an accuracy of $82\%$. These results show knot tying is sensitive to discretization as very small changes in the actions can lead to different topological states. As the space of available discretized actions is already rather large (larger action spaces would make planning even more difficult) we conclude that discretization of the action space is not a suitable approach to the knot-tying problem.

### 5.2 Success Rate of Different Algorithms

In this experiment, we compare several algorithms, including TWISTED and its ablations.

**Low level only:** We modify TWISTED to **not use** any high-level information. Essentially, using random search over configurations with curves as actions. As there is no notion of topological states, there is no way to use the inverse model here. Instead, we sample *random curves*. It is important to notice that for this baseline, the search does not get feedback along the trajectory (in TWISTED, we do, for instance, count the number of crosses). We use this baseline to demonstrate how crucial high-level information is for knot tying.

**Low+high level:** We modified TWISTED not to use the inverse model. Instead, we sample random actions replacing those suggested by the inverse model. Unlike the previous baseline, we do try to follow a high-level plan. This baseline demonstrates the trade-off between intensive but more accurate action prediction (inverse model) vs. an approach of guessing many random actions and seeing if any suffice.

**SAC+HER:** In this baseline, we learn a stochastic policy using the Soft Actor-Critic (SAC)[41], with Hindsight Experience Replay (HER)[42], and after training we replace our inverse model with the policy. The objective of this baseline is to establish the performance of model-free RL methods and the challenging problem of knot-tying.

**TWISTED, RND:** TWISTED using the *random* heuristic for topological state selection.

**TWISTED, CRS:** TWISTED using the *prioritize-crosses* heuristic for topological state selection.

**Evaluation protocol:** To evaluate the performance on different difficulty levels, we split our collected data $D$ (Section 4.5) into three levels: `easy`, `medium`, and `hard`. To classify the problems (topological goal states), we counted the frequency each topological state has been recorded. `Easy`,

`medium` and `hard` are the 33%, 66%, and 100% percentiles appearing the most in the data. From every class of problems, we sampled ten representatives.

**Results:** None of the algorithms solve `medium` or `hard` in the time limit of 1800 seconds, demonstrating the hardness of the knot-tying problem. Figure 3 (a) shows the number of solved tasks vs. the running time for `easy` problems. First, observe that "low level only" is barely able to solve two out of the ten problems. This validates our earlier hypothesis in Section 4.5 that the problem is too sparse to solve without prior knowledge of the problem structure (such as our high-level search). Surprisingly, the model-free RL baseline is barely better than the random search. We observed that during training, it did manage to consistently solve all 1-cross problems, but already for 2-cross problems success rate was near zero. This suggests that knot-tying is a hard task to learn end-to-end without proper domain knowledge. We hypothesize that the main reason this baseline fails is due to the discrete nature of the topological states – in such cases algorithms cannot generalize between "similar" states because as categorical variables, there is no notion of similarity, only the relation of equality. Even a well utilized exploration method such as HER does little to mitigate this problem, because it can only reinforce patterns for goals we reach, and as we saw when acting randomly, like RL agents do at the beginning of training, there is little chance to advance to topological states with many crosses (see Section 4.5). Finally, regarding the baselines, we see that because "low + high level" is so inferior to the full TWISTED versions, the inverse model is a valuable component of our full solution. Comparing "TWISTED, RND" and "TWISTED, CRS", we observe that the results are not conclusive. To identify the better model, we sampled 15 additional `easy` goals to get a statically significant separation on which is the better variant of TWISTED. The "TWISTED, CRS" solved a total of $24/25$, and the "TWISTED, RND" solved only $19/25$. Under a Z-test the "TWISTED, CRS" has a statistical significance of being better than the "TWISTED, RND" (using $\alpha$-level of 0.05), showing that planning deeper and utilizing prior knowledge (number of crosses) is preferable. Therefore, in our next experiments, we use the "TWISTED, CRS" version.

### 5.3 Sensitivity Analysis
To motivate the usage of TWISTED in real-world applications we test what happens if the model of the world in the planning computation *is mismatched with the evaluation environment*. In our experiments, we focus on the friction coefficient. First, we validate that friction indeed significantly impacts rope tying. To measure this, we compare the next topological state observed when applying the same action from the same low-level state, under different friction coefficients. We evaluated over 600,000 actions, and only $82\%$ curves had the same topological state as the original friction value. Next, we evaluate the performance of TWISTED trained with a single friction coefficient on simulated environments with different frictions of the rope.

**Variants:** $100\%$ friction denotes the default Mujoco friction, and the one we use for TWISTED. The $95\%$ and $105\%$ variants, denote decreasing and increasing the friction by $5\%$.

**Results:** The performance of TWISTED is well-maintained with the different friction coefficients (Figure 3 (b)). This asserts that TWISTED can handle some variations in the environment's physics such as friction (even though the resulting trajectories might be different than the original ones).

### 5.4 Generalization to Unseen Topological States
The number of available topological states for states with 3 or 4 crosses is above 500 and almost 8000 correspondingly. Naturally our data does not contains all of them because it is hard to sample topological states of higher crosses (see Section 4.5 for analysis). Thus, we require our algorithm to handle unseen topological states. We evaluate whether TWISTED can tie knots where the goal state was not represented in $D$. For this, we take topological states with *three* crosses not seen in $D$, and topological states with *four* crosses. In these out-of-distribution cases we expect the inverse model to contribute less than in well represented states, and we expect that the planning components in TWISTED will compensate for this distribution shift. For this reason, we extend search time by a factor of $4\times$. Results are shown in figure 3 (c). Figure 3 (c) shows that TWISTED solved two out of eight problems with unseen three cross states. Those states are harder to reach because they were never seen in $D$ during data collection. In figure 3 (c) we see that TWISTED solved three out of ten problems with unseen states with four crosses. We recall that our data contain only one, two, and

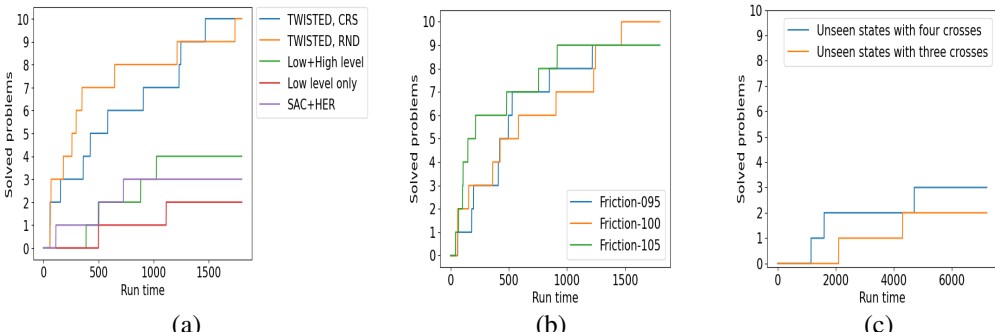

Figure 3: Anytime Success Rates for Different Settings. The X-axis is the algorithm run time and the Y-axis is the number of solved problems. (a) Differnt knot tying algorithms, (b) "TWISTED, CRS" with different ropes and different friction to knot tying, and (c) "TWISTED, CRS" on unseen states with three crosses and four crosses, with eight and ten problems respectively

| Algorithm | Solved problems | Friction | Solved problems | Goal states | Solved problems |
|---|---|---|---|---|---|
| Low level only | 2 | 95% | 9 | 3 crosses | 3/10 |
| Low+high level | 4 | 100% | 10 | 4 crosses | 2/8 |
| SAC+HER | 3 | 105% | 9 | | |
| TWISTED, RND | 10 | | | | |
| TWISTED, CRS | 10 | | | | |

Table 1: **Experiments Summary:** (a) Success Rate of Different Algorithms, (b) Sensitivity Analysis, and (c) Generalization to Unseen Topological States

three crosses, and therefore these results show that TWISTED is not only memorizing the data, but can generalize to some degree to unseen goal states.

## 6 Limitations

Our work has several important limitations that need to be addressed in order to make it more practical and useful in real-world applications. First, simulation accelerates the training process but introduces a *sim2real gap* between the simulation and real-world performance. This gap should be tested on a real robot using real ropes. Furthermore, in this work *we also simplified the problem*; we control a free-moving end-effector instead of controlling a manipulator (which might make some curves unfeasible in some scenarios), and we get a perfect representation of the rope, where in reality we would need first to estimate one. Finally, as our experiments demonstrate, TWISTED has shown better performance on frequent data from the `easy` problems, but its *performance decreases when trying to solve more rare or unseen goals from* `medium` *and* `hard`.

## 7 Outlook

We presented TWISTED – a hierarchical planning algorithm for knot tying, that relies on knot-theory and a learned inverse model to automatically solve problems that previously required access to human demonstrations. TWISTED outperforms various baselines, including a model-free deep RL agent, and we demonstrated robustness to simulation parameters such as friction, and generalization to problems not seen during training (even to problems of greater complexity). We see this as an important step towards general 1D object manipulation, and to the best of our knowledge, this is the first work that manages to tie knots using random data instead of demonstrations.

An exciting area for improvement would be to utilize TWISTED as a demonstrations provider to generate "valuable" data for an off-policy RL algorithm, either by distilling the planner into a policy [43] or by combining RL with imitation learning [44]. This could be the missing prior knowledge that RL methods lack for knot-tying tasks (cf. Section 5). A different interesting future direction might be to improve the data collection process to *seek* novel states instead of relying on random actions. Collecting data from states where the system is less capable, could ultimately provide data of higher quality and improve the performance of our learned inverse model. Finally, it would be interesting to test TWISTED on a real system overcoming challenges such as estimating the rope configuration and executing precise rope manipulation actions.

## 8 Acknowledgement

Supported by a grant from the Israeli Planning and Budgeting Committee.

This work received funding from the European Union (ERC, Bayes-RL, Project Number 101041250). Views and opinions expressed are however those of the author(s) only and do not necessarily reflect those of the European Union or the European Research Council Executive Agency. Neither the European Union nor the granting authority can be held responsible for them.

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

# 9 Appendix

## 9.1 P-data, an abstract representation for rope states

A common way to abstract the state space in knot tying is using the **P-data** representation [24]. The P-data representation translates a rope configuration to a matrix of discrete values that depends on the number of link intersections. The P-data algorithm stages are: (1) project the 3D rope onto a 2D on the horizontal plane. (2) Select rope direction by defining the head and tail of the rope. (3) Move from head to tail and count the number of intersections along the path, starting from 1 to N. Those intersections are also called crosses. Finally, (4) each intersection gets **over/under** value based on which segment is over the other in the height dimension and also gets a **sign** plus/minus. The sign defined as

$$sign = \frac{\overrightarrow{l}_{over} \times \overrightarrow{l}_{under}}{|\overrightarrow{l}_{over} \times \overrightarrow{l}_{under}|} \cdot \overrightarrow{e_z},$$

where $e_z$ is the unit normal of the horizontal plane, and $l_{over}$ and $l_{under}$ are the two strands directional vectors. Examples of P-data of projected knots in Figure 1.

## 9.2 Reidemeister Moves

The various Reidemeister Moves are depicted in figure 4.

## 9.3 Collection process

Our data collection process is split into two steps, the first is random sampling with resets and the second is noisy re-sampling.

The *random sampling with resets* works as follows: We maintain a set of configurations we have already seen during data collection and their respective number of crosses, i.e. $D_Q = \{(q, \texttt{Cross}(\texttt{Top}(q))\}_{q \in D}$. For every data collection iteration $t$, we load the simulation with a configuration sampled uniformly from $q_t \sim D_Q$, take a random 100 curves $c_t^i$ (for $i \in [1 \ldots 100]$), and reach a new configuration $q_{t+1}^i$ with a topological state $s_{t+1}^i$. We sample the curve parameters uniformly: the link is discrete and sampled from [1, 21]. The other three, $x$, $y$, and $z_{max}$, are continuous variables sampled from [-0.5, 0.5], [-0.5, 0.5], and [0.001, 0.07], respectively. If the number of crosses in $\texttt{Cross}(s_{t+1}^i) > \texttt{Cross}(s_t)$ the transition $(q_t, s_t, c_t^i, q_{t+1}^i, s_{t+1}^i)$ is added to $D$, and the configuration $q_{t+1}^i$ is added to $D_Q$.

In *noisy re-sampling*, the goal is to increase the amount of data using previously collected data. First, we sample transitions from our data set (thus they are "interesting", i.e. move the agent to higher number of crosses), add noise to the action, and obtain a new transition based on the same starting configuration and the modified action. The noise distribution is uniform and characterized by four parameters that offset the current action: The link offset is sampled from $\{-1, 1\}$. The $x$, $y$, and $z_{max}$ offsets are sampled from [-0.05, 0.05] but are clipped to stay within the limits of [0.001, 0.07] for $z_{max}$ and [-0.5, 0.5] for $x$ and $y$. See Algorithm 2 for more information.

## 9.4 Inverse model

Our inverse model architecture is detailed in figure 5.

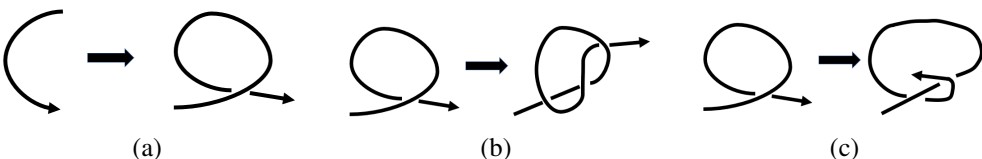

(a)  (b)  (c)

Figure 4: (a) Reidemeister move one, (b) Reidemeister move two and (c) cross.

**Algorithm 2** Data Collection

```
 1: while time < TimeBudgetStageOne do
 2:     state = GetStateFromData()                          ▷ Select state randomly
 3:     PotentialActions = SampleRandomAction()
 4:     for action in PotentialActions do
 5:         NextState = RunActionInSim(action)
 6:         if ValidaStete(NextState) then    ▷ Checks if the action increases number of crosses
 7:             SaveAction(action)
 8:         else
 9:             ResetState()
10:         end if
11:     end for
12: end while
13: while time < TimeBudgetStageTwo do
14:     state, action = GetStateAndActionFromData()          ▷ Select tuple randomly
15:     noise = SampleActionNoise()
16:     NewAction = action + noise
17:     NextState = RunActionInSim(NewAction)
18:     if ValidStete(NextState) then       ▷ Checks if the action increases number of crosses
19:         SaveAction(action)
20:     else
21:         ResetState()
22:     end if
23: end while
```

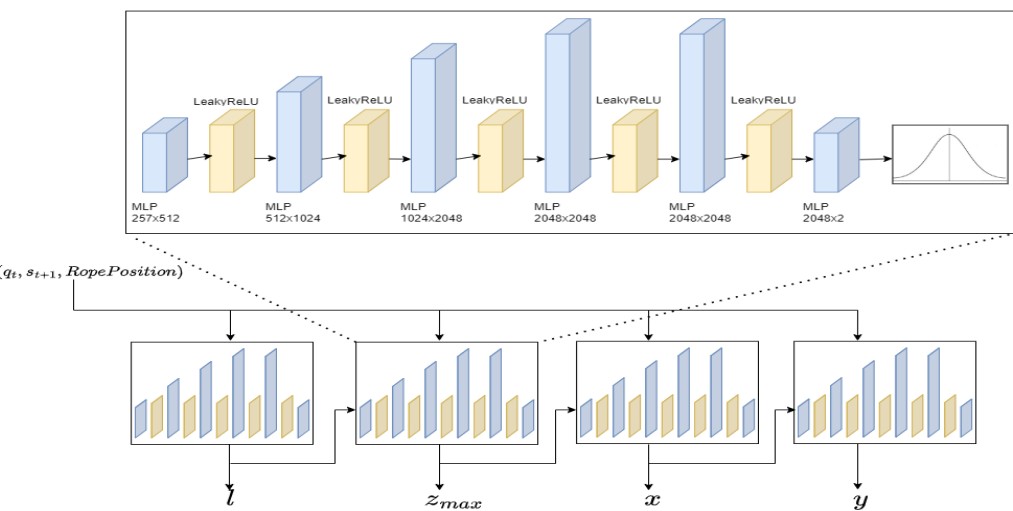

Figure 5: Inverse model - Auto-regressive Stochastic Network. The network predicts an action in an auto-regressive manner: first is predicts the link index $l \in [1, L]$, then the height of the curve $z_{max}$, finally it predicts the $x$ and $y$ coordinates of the curve. All predictions are stochastic (Multinominal for link index, and Gaussian otherwise). Besides the previous elements, the input of each element includes the current configuration $q_t$, the next topological state $s_{t+1}$, and the link positions of all the rope links. The weights of the sub-components are not shared.

