# OpenReview forum: "Hierarchical Planning for Rope Manipulation using Knot Theory and a Learned Inverse Model"
_robot-learning.org/CoRL/2023/Conference — CoRL 2023 Poster_

### Official Review · Reviewer_6Lwg · 2023-07-13

**Confidence:** 4
**Originality:** Good
**Technical Quality:** Good
**Clarity Of Presentation:** Good
**Impact:** 3

**Recommendation:**

Weak Accept: I recommend accepting the paper, but will not argue for my recommendation if the majority of other reviewers have a different opinion.

**Review:**

Strengths
- Knot tying is a very challenging task and the results shown in the video are truly impressive. The paper definitely makes a step forward in the field of rope manipulation.
- The proposed hierarchical planning method seems really promising. It transforms the high-dimensional state space of rope into a fixed set of topological states based on the know theory, which significantly improves the efficiency of planning. The low-level inverse model is trained in a self-supervised manner and also works very well in tracking the high-level plan.
- The paper conducts thorough experiments to evaluate the effects of different design choices. Although no physical experiments is conducted, the authors show in simulation that the proposed method is robust to different phyiscal properties.

Weaknesses
- The proposed method is only tested in simulation but not real world.
- The paper makes a lot of simplifications in terms of perceptions and controls. It assumes perfect state estimation, which is very difficult in real world due to self-oclcusion. Also, it uses a floating gripper that can move freely. While in real world, a robot arm will introduce additional topological constraints that makes certain actions infeasible. Sometimes, the robot may need to release the gripper and regrasp the rope from the other side.

**Quality Of The Limitations Section:**

Limitations are addressed clearly

**Questions For Rebuttal:**

- In line 113, it says "The known reachable configurations, is a tree T", while in line 127, it says "identifies the reachable topological states in the graph". Can you clarify whether the known reachable configurations is stored as a tree or a graph?
- Fig 1 is a bit confusing. In the bottom left,  topological state is represented by grey sphere, while in the rest of the figure, grey cube is used.
- Why do we need to store all known reachable configurations? Is it because the transition function is not well defined under the topological representation and Reidemeister moves? If so, can I see it as some sort of probabilistic roadmap, where you build a roadmap (graph) made of topological states, and then use BFS to find high level plan at test time?
- During planning, multiple (K=6) curves are generated by the inverse model (line 163). I'm wondering how do you determine which curve leads to the desired topological state s'.
- How does the Low level only (line 223) work? Is it basically a random policy that samples in the low-level action space? If so, it doesn't seem to be a fair comprison. To show the advantage of hierarchical planning, you compare with low-level planner such as CEM or MPPI.
- The ablation shows that a discretized actions space (discretized curve) hurts performance, and a continuous action is inferred by the inverse model. But I also notice that the picked point selection is discretized, i.e., rope is divided into multiple links. I wonder how much does this affect the performance. And how does the performance change with respect to the number of links L

**Robotics Focus:**

Highly relevant to robotics but no hardware experiments

**Summary Of Paper:**

The paper propose a hierarchical planning algorithm for knot tying, which is built on knot-theory and doesn't require access to human demonstrations. The proposed algorithm first discretize the continuous state space of rope into topological states, and plan in this discretized space using graph search. To move between the configurations in the high-level plan, the authors utilize a learned inverse model, which predicts pick position and motion autoregressively. The authors show in simulation that the proposed method outperform various baselines.

**Summary Of Recommendation:**

Overall, I think it's a good paper that makes solid contribution in rope manipulation. The authors leverage knot theory to design a hierarchical planning method that tackles a really challenging task robotic manipulation task, knot tying. Although no physical experiment is performed, I think it's understandable since the focus is mainly on the planning side. And I do not expect all challenges (perception, planning, controls) are solved in a single paper. Therefore, I recommend weak accept and will raise my rating if all questions are clarified.

---

> ### Author Response · Authors · 2023-08-09
> **Authors response**
>
> We thank the reviewer for their helpful comments:
>
> * “The paper makes a lot of simplifications in terms of perceptions and controls. It assumes perfect state estimation, which is very difficult in real world due to self-oclcusion. Also, it uses a floating gripper that can move freely. While in real world, a robot arm will introduce additional topological constraints that makes certain actions infeasible. Sometimes, the robot may need to release the gripper and regrasp the rope from the other side.”
>
> Response: As our paper demonstrates knot-tying even under our simplified assumptions, is very difficult. We think that by isolating the perception and control aspects of this problem, we can progress toward better planning algorithms for this problem. Of course that to deploy such a system in the real world, those problems should be addressed. We leave those for future work.
>
> * “During planning, multiple (K=6) curves are generated by the inverse model (line 163). I'm wondering how do you determine which curve leads to the desired topological state s'.”
>
> Response: We test in simulation, applying the action and observing the next topological state.
>
> * "How does the Low level only (line 223) work? Is it basically a random policy that samples in the low-level action space? If so, it doesn't seem to be a fair comprison. To show the advantage of hierarchical planning, you compare with low-level planner such as CEM or MPPI."
>
> Response: Because the reward in this task is binary, there is no learning \ optimization signal before a single successful trajectory is found. Thus, a random policy is as good as any other method for finding the first success. But after finding this initial guess, the task of reaching the next topological state is already solved, thus the benefits of those methods will not come to fruition, and they would perform just as well as a purely random search.
>
> * "The ablation shows that a discretized actions space (discretized curve) hurts performance, and a continuous action is inferred by the inverse model. But I also notice that the picked point selection is discretized, i.e., rope is divided into multiple links. I wonder how much does this affect the performance. And how does the performance change with respect to the number of links L”
>
> Response: This is a great suggestion that we’ll execute. Due to time restrictions, we cannot complete this during the rebuttal time window.
>
> * "Why do we need to store all known reachable configurations? Is it because the transition function is not well defined under the topological representation and Reidemeister moves? If so, can I see it as some sort of probabilistic roadmap, where you build a roadmap (graph) made of topological states, and then use BFS to find high-level plan at test time?”
>
> Response: We store all the configurations because given a topological state it is unknown how to find reachable configurations for that state. Instead, we expand new configurations (indeed, similar to PRM) and we make sure that we only keep transitions between configurations that correspond to advancing the number of crosses in the topological states.
>
> * "In line 113, it says "The known reachable configurations, is a tree T", while in line 127, it says "identifies the reachable topological states in the graph". Can you clarify whether the known reachable configurations are stored as a tree or a graph?”
>
> Response: We apologize for the discrepancy, we will fix it to Tree along the Paper.
>
> * All text and presentation concerns will be addressed in the final version.

---

### Official Review · Reviewer_fZRK · 2023-07-13

**Confidence:** 3
**Originality:** Good
**Technical Quality:** Good
**Clarity Of Presentation:** Good
**Impact:** 3

**Recommendation:**

Weak Accept: I recommend accepting the paper, but will not argue for my recommendation if the majority of other reviewers have a different opinion.

**Review:**

The main idea of this paper is interesting and the method is supported with empirical results. Admittedly, I am not familiar enough with the rope tying literature to know if something similar has been attempted before.

The execution of the paper should be improved.  There should be more empirical data especially since this paper is simulation only. The presentation of the paper is unsatisfactory. Please refer to the “Questions for rebuttal” section for more details.

Strengths:

1.	Clear proposed idea

2.	Favorable empirical results

Weaknesses:

1.	Limited empirical data

2.	Unpolished presentation

3.	No hardware experiments

**Quality Of The Limitations Section:**

Limitations are addressed clearly

**Questions For Rebuttal:**

1.	The sensitivity analysis under different friction coefficients needs to be justified further. The authors did not state how much the friction coefficient was changed to yield “82% curve has the same topological state as the original friction value.” The difference between the results in figure 3(b) differ by up to 2 trials at times, which does yield a 20% success rate difference. More trials are needed to support the sensitivity claim, i.e. improve similarity to significantly over 82%.

2.	What exactly is the setup for the SAC+HER baseline? The text says “after training we replace our inverse model with the policy.” That sounds like the high-level planner is still used. If so, doesn’t that just mean SAC+HER is significantly worse than the low-level action generator?

3.	I suggest moving the TWISTED Algorithm from the appendix to the main text. It was very useful for understanding the algorithm in my opinion.

4.	The purpose of SelectTopologicalState is unclear to me, given that the rope is starting from a specific $q_{init}$. Is this just finding the next $s$ for the rope to goto? i.e. the full topological state trajectory of the rope should be $s_{init}, s_{selected}, …$

5.	“FollowPlan” and “RandomExpand” in the appendix are not defined.

6.	Should “FollowPlan” take in the current configuration of the rope as well? How many topological states is “FollowPlan” expected to traverse? If the plan following fails, for instance the rope did not end up in the next topological state in $P_{selected}$, is there any replanning?

7.	In the abstract “…at the high level, use ideas from knot-theory to plan a sequence of rope configurations…”. Shouldn’t “rope configurations” be “topological states”? Also there should not be a hyphen in “knot-theory.”

8.	Section 4.2 Data structures: is the known configurations really a tree? The “reachable configurations” in Figure 1 contains cycles.

9.	Section 4.3 “The result of the BFS is a set of paths” Could you clarify how the BFS may return multiple paths?

10.	On page 4 “SelectTopologicalState(): identifies…in the graph.” Is “graph” referring to the tree T?

11.	Misaligned “S1” in the “Hi-level planner” box in figure 1.

12.	On page 4 “our aim is to increase the frequency in which topological states of higher complexity are utilized as the initial state in the high-level plan”. What is the reason behind this?

13.	Should “low-level states” in the caption of figure 1 be “rope configuration”?

14.	I suggest providing an animated visualization of the TWISTED algorithm. It would be helpful for the reader.

15.	The texts in Figure 3 need to be larger.

**Robotics Focus:**

Relevant but unlikely to deploy to hardware in near future

**Summary Of Paper:**

This paper presents TWISTED, a pipeline for tying knots without demonstration through hierarchical planning. The high-level planner plans a path with breadth first search in the space of topological states, and the low-level planner uses a learned auto-regressive model to transition between topological states. The proposed method is evaluated in simulation using Mujoco and outperformed the baselines. TWISTED was tested on different friction coefficient and yielded similar results.

**Summary Of Recommendation:**

Overall, I think the paper is quite interesting and the proposed idea could have impact. I am not familiar enough with rope manipulation to judge exactly how this paper stands relative to the literature, but it seems like a decent advancement. This could make a solid paper with improvements in execution and the presentation issues ironed out. However, in its current form, I cannot recommend it for CoRL.

---

> ### Author Response · Authors · 2023-08-09
> **Authors response**
>
> We thank the reviewer for their helpful comments:
>
> * “The sensitivity analysis under different friction coefficients needs to be justified further. The authors did not state how much the friction coefficient was changed to yield “82% curve has the same topological state as the original friction value.” The difference between the results in figure 3(b) differ by up to 2 trials at times, which does yield a 20% success rate difference. More trials are needed to support the sensitivity claim, i.e. improve similarity to significantly over 82%.”
>
> Response: Manipulating deformable objects requires very accurate actions, and even small changes in the action can move the rope to a different topological state. In experiment 2, we changed the friction by 5%, and as a result, 18% of the state-action pairs in our dataset led to different topological states than using the original friction. Note that it does NOT change the performance of TWISTED by much, because our planning component appears to be robust to these changes.
> The goal of this experiment was not to test how much the dynamics change, but rather the effect it has on the general performance of TWISTED.
> We thank the reviewer for the comment and will elaborate on it more.
>
> * “What exactly is the setup for the SAC+HER baseline? The text says “after training we replace our inverse model with the policy.” That sounds like the high-level planner is still used. If so, doesn’t that just mean SAC+HER is significantly worse than the low-level action generator?”
>
> Response: Correct, the high-level planner is the same as in TWISTED and we replace the inverse model with SAC+HER. The motivation is to give SAC the high-level information that TWISTED utilizes. Without using the high-level planner, this task would become too sparse for SAC to solve.
>
> * “The purpose of SelectTopologicalState is unclear to me, given that the rope is starting from a specific q_init. Is this just finding the next S for the rope to goto? i.e. the full topological state trajectory of the rope should be S_init, S_selected…”
>
> Response: After a few iterations of TWISTED some topological states were already discovered. Given this state of the search algorithm and the goal topological state, the SelectTopologicalState is responsible for selecting a reachable topological state we would like to resume planning from. The motivation is to “cluster” configurations according to their topological states, and thus bias the search towards configurations of harder to reach.
>
> * “Should “FollowPlan” take in the current configuration of the rope as well? How many topological states is “FollowPlan” expected to traverse? If the plan following fails, for instance the rope did not end up in the next topological state in P_selected, is there any replanning?”
>
> Response: The “FollowPlan” gets the current configuration and high-level plan and traverses the topological states in the high-level plan. If it fails to traverse one of those edges, the method returns to allow a new iteration of TWISTED to start.
>
> * “Section 4.2 Data structures: is the known configurations really a tree? The “reachable configurations” in Figure 1 contains cycles.”
>
> Response: To represent the reachable configurations we are using a directed tree. See general comments above. Re cycles: the tree does not contain cycles, because we add an edge from a configuration q_1 to configuration q_2, only if the number of crosses in q_2 is greater than q_1, thus the no cycle property is maintained. We will make the text clearer around this point. We’ll make sure the figure does not contain cycles.
>
> * “Section 4.3 “The result of the BFS is a set of paths” Could you clarify how the BFS may return multiple paths?”
>
> Response: The high-level planner does not stop after finding one plan. Instead, we limit the depth of the search by the number of crosses of the goal state and we return all the plans from the initial state to the goal state found by the BFS search.
>
> * "In the abstract “…at the high level, use ideas from knot-theory to plan a sequence of rope configurations…”. Shouldn’t “rope configurations” be “topological states”? Also there should not be a hyphen in “knot-theory.”
>
> Response: Yes, we will change both.
>
> * “On page 4 “our aim is to increase the frequency in which topological states of higher complexity are utilized as the initial state in the high-level plan”. What is the reason behind this?”
>
> Response: The higher the number of crosses the more difficult it is to find a suitable action. Biasing in the above-mentioned way allows us to expand the search from more interesting topological states more often.
>
> * “I suggest providing an animated visualization of the TWISTED algorithm. It would be helpful for the reader.”
>
> Response: We are working on a video that will illustrate the entire algorithm.
>
> * All text and presentation concerns will be addressed in the final version. We will also clarify the section regarding the method.

---

### Official Review · Reviewer_AJzJ · 2023-07-15

**Confidence:** 3
**Originality:** Very Good
**Technical Quality:** Very Good
**Clarity Of Presentation:** Fair
**Impact:** 4

**Recommendation:**

Strong Accept: I recommend accepting the paper and will argue for my recommendation even if other reviewers hold a different opinion.

**Review:**

This is an important problem. Enabling robots to manipulate ropes and wires at the same level as a human being would be very valuable commercially, and would greatly increase our understanding of how to solve more general dextrous manipulation problems. This is particularly interesting because, unlike many manipulation problems involving rigid physics, ropes present a continuum of degrees of freedom, and appropriate understanding needs to be developed to handle this.

I am not familiar with knot theory - and, I suspect, most readers will not be as well. Therefore, a more comprehensive review would have been helpful, in my opinion the background is too short. For example, the authors introduce the notion of "Reidemeister moves" which is an important notion because each such move represents steps in the high-level planner, but don't attempt to offer an explanation. At least something needs to be said about what those moves represent intuitively and why they are the appropriate knot-theoretic notion to use for high-level planning steps. If the authors think this is too complex to define properly - and I am skeptical this is the case - they should instead explain why this is too complex.

The writing is wordy, which makes certain sections feel like they take up a lot of space without containing a lot of content. The "prior work" section is arguably the worst offender. There is no need to repeatedly explain why the other papers in the literature are different from this one in point-by-point detail - this is a paper, not a review rebuttal. If this kind of detail is offered at all, this should be presented in the conclusion section after the reader has read about the paper's techniques. This section should therefore be cut to 1/3 its current length and the extra space used to better explain the background so that readers have the tools they need to understand the differences between this paper and other papers on their own.
* For example, the "knot un-typing paragraph" could be replaced with "[these authors] have examined the task of un-tying a knot in a given rope, with most approaches operating using high-level abstractions which mirror the ones we study". There is no need to say that bringing a rope to an arbitrary state - rather than to an untied state - is more complex, as this is obvious.

Section 4 could have a more informative title than "Method", which instead describes something about how the method works.

Section 4 focuses on the steps taken by the planning algorithm, which is a tree-based algorithm where nodes are added based on picking a random node and executing random actions. One could easily generalize this to more general tree search algorithms, which might be faster at the cost of being more complex or more difficult to implement. It would be nice to formulate this in a general way, and say that a particular way of searching the tree is used by the authors because it works.

The data collection scheme is not described, but is critical to the algorithm's ultimate success. This part is therefore important enough that it needs to be described in the main body, not the appendix.

Figure 3 fonts are too small, and because my eyesight is not great I cannot tell what algorithms are doing well and which ones are not. Please make this figure significantly larger, so that its fonts are at least the same size as the paper's text.

In the text, "Figure 3" is not capitalized consistently. Please make this consistent.

The CoRL-required "limitations" section is not used effectively, because the authors use it to describe limitations a second time that have already been stated earlier in the text. As mentioned earlier, this paper would be significantly better if more space was dedicated to describing the background and methods in more detail rather than moving critical information into the appendix, which means that repeating information here harms the paper. Please consider moving the limitations to a subsection of the methods section, and consider moving other parts of the text that describe limitations elsewhere into this section.
* Additionally, the most important limitation of TWISTED is that - while a better algorithm than baselines - it also does not solve in the medium or hard levels. This should be mentioned in limitations, so that readers who don't read the rest of the paper can quickly see that this is the level of performance to be expected. Note that being upfront about this limitation does not devalue the work, since knot-tying is an important enough problem that even limited performance improvements are of good scientific value.

**Quality Of The Limitations Section:**

Limitations are addressed clearly

**Questions For Rebuttal:**

Why are knot-theoretic notions which are critically important to understanding this work not defined in the main text, whether intuitively or rigorously, and instead relegated to references and/or appendix?

Does the tree search algorithm you use - which, if I'm not mistaken, consists of picking a random node in the tree, and executing random actions out of that node - have a standard name?

I presume Fig. 3(b) shows different kinds of rope with the same algorithm - which algorithm is used here?

Are there any key difficulties or limitations involving the data-collection scheme of Section 4.5 or other parts of the paper that have not been described in detail in the main body?

**Robotics Focus:**

Highly relevant to robotics but no hardware experiments

**Summary Of Paper:**

This paper proposes a pipeline for making it possible for a robot to tie knots and manipulate wires without relying on a large set of human demonstrations. This is done by splitting the approach into (i) a low-level part which involves learning how to represent states of the rope from perceptual data, and (ii) a high-level part that uses ideas from knot theory to reason about how the rope needs to be manipulated to bring it to the desired state. The approach is shown to generalize, demonstrating the ability to tie knots that were not seen during training. In total, the approach yields a system capable of handling knots in a category the authors label "easy" - a significant improvement on baselines such as deep RL, which can't even do that - in my view, fantastic progress.

**Summary Of Recommendation:**

This paper brings great improvements on a hard problem, using an approach with a significant amount of bells and whistles which are largely justified and make sense. I am happy with the results and evaluations. The main downside is that the paper does not describe enough detail about the setting or method in the main body, due to a combination of repetitive text, too much detail about prior work, and generally inefficient use of space. The paper would be significantly improved with a a rework of the writing, but I think is already interesting enough that I am more than happy to see it published now in spite of its flaws.
* Since important parts of the algorithm such as the data-collection scheme of Section 4.5 are not described at all in the main text, it's also possible there are hidden problems that should have been mentioned as limitations have been withheld. I didn't detect such issues, but am very curious if other referees might have.

---

> ### Author Response · Authors · 2023-08-09
> **Authors response**
>
> We thank the reviewer for their helpful comments:
>
> * “The data collection scheme is not described, but is critical to the algorithm's ultimate success. This part is therefore important enough that it needs to be described in the main body, not the appendix.”
>
> Response: We will extend and move this section to the main body
>
> * “Why are knot-theoretic notions which are critically important to understanding this work not defined in the main text, whether intuitively or rigorously, and instead relegated to references and/or appendix?”
>
> Response: We will make the knot theory background more accessible.
>
> * “Does the tree search algorithm you use - which, if I'm not mistaken, consists of picking a random node in the tree, and executing random actions out of that node - have a standard name?”
>
> Response: We emphasize that our algorithm does not pick a random configuration, rather it prioritizes configurations according to the associated topological state (i.e. not random). In “TWISTED, CRS” we prefer topological states with more crosses as a heuristic, and in “TWISTED, RND” we choose a random topological state. Once a topological state is selected, a reachable configuration belonging to that topological state is sampled.
>
> * “Are there any key difficulties or limitations involving the data-collection scheme of Section 4.5 or other parts of the paper that have not been described in detail in the main body?”
>
> Response: As section 4.5 states, it is very hard to efficiently collect data, therefore, at a high level we used a mixture of random data collection with resets, and we also sampled around previously discovered transitions that advance the number of crosses.
> More specifically, given a description of what constitutes an interesting transition (and a method to identify those), we apply the following and only save interesting transitions into our data set:
> 1. We continuously sample random curves in the form of episodes: we start in the initial configuration (that is unique) and we try to find new interesting transitions K=100 times. If we find such a transition, we try to expand it for K=100 times, and so on. Once an episode terminates, we go back to the initial rope configuration and repeat the process.
> 2. We sample transitions from our data set (thus they are interesting), add noise to the action, and obtain a new transition based on the same starting configuration and the noisy action.
> We will make this more explicit in the text.
>
> * “Section 4 could have a more informative title than "Method", which instead describes something about how the method works.”
>
> Response: We will change it to TWISTED.
>
> * "I presume Fig. 3(b) shows different kinds of rope with the same algorithm - which algorithm is used here?”
>
> Response: At the end of “Success Rate of Different Algorithms” results section we indicate “TWISTED, CRS” “Therefore, in our next experiments, we use the “TWISTED, CRS” version.

---

> > ### Comment · Reviewer_AJzJ · 2023-08-15
> > **Acknowledgment**
> >
> > Thank you very much for your rebuttal. I don't have any further questions, and am interested in discussing about what other referees thought of this work.

---

### Official Review · Reviewer_o22m · 2023-07-21

**Confidence:** 4
**Originality:** Good
**Technical Quality:** Good
**Clarity Of Presentation:** Fair
**Impact:** 3

**Recommendation:**

Weak Reject: I recommend rejecting the paper, but will not argue for my recommendation if the majority of other reviewers have a different opinion.

**Review:**

Strengths:

- The authors present a novel approach to rope manipulation that is compelling.

Weaknesses

- The experiments the authors have chosen to highlight their proposed method and the research questions they proposed seem disconnected.
- The authors do not validate their method against a competitive state-of-the-art background. I would suggest applying a hierarchical RL method.
- The presentation of the results within the paper is poorly executed. A table would help a lot. Figure 3 is not referenced in the correct places and is not intuitive to follow (i.e. what is the y-axis).
- The paper needs to be thoroughly edited to rectify writing issues.
- Sim-to-real is a significant overheard for transferring this method to a real-world environment although the authors allude to "real-world applications" in their sensitivity analysis as if overcoming small changes in friction make it likely that this will naively transfer.
- The format of the related work induces a lot of repetition.

**Quality Of The Limitations Section:**

Limitations are addressed clearly

**Questions For Rebuttal:**

- The paper needs to be thoroughly edited to rectify issues, for example, figure 2 isn't referenced in text (I do not believe); Figure 1 is on page 2 but not referenced until page 4; some non-intuitive phrases in lines 61-21, 59-65, 122-124.
- What is the notation q supposed to represent? It is not clear in text.
- Acronyms are used but not spelt out, i.e. BFS
- What is meant by "The difficulty in manipulating ropes arises from the limited number of actions to change the rope's topological state"?
- Is rope a 1-dimensional object? Is it not simulated as a 3D object with many degrees of freedom?
- What is the inverse model based on? I would suggest adding the network architecture diagram into the text as this would be helpful for a reader trying to follow the text.


**Robotics Focus:**

Highly relevant to robotics but no hardware experiments

**Summary Of Paper:**

The authors present a hierarchical approach to manipulating rope into a desired goal configuration. The hierarchy is broken down into a high-level solver with a discrete representation of the rope and a lower-level that uses a neural network to plan a sequence of robot actions between subgoals dictated by the high-level solver. The authors present several simulated experiments that investigate their approach against some simple baselines.

**Summary Of Recommendation:**

The authors propose a compelling method to complete rope manipulation tasks. However, the clarity of presentation and the choice of experiments along with baselines needs to be addressed for the paper to show the advantages of their approach over other competitive methods.

---

> ### Author Response · Authors · 2023-08-09
> **Authors response**
>
> We thank the reviewer for their helpful comments:
>
> * “The authors do not validate their method against a competitive state-of-the-art background. I would suggest applying a hierarchical RL method.”.
>
> Response: Thank you for the suggestion to include HRL in our experiments, however, we think that HRL would not be competitive since it would have to learn the high-level abstractions from scratch while TWISTED has this information integrated as prior knowledge. Our SAC + HER experiment demonstrates that even a SOTA RL algorithm integrated with high-level planning is still unable to make meaningful progress toward solving this task. That being said, if the reviewers have suggestions for specific HRL algorithms that they believe will be competitive we would be happy to run them.
>
> * "What is the notation q supposed to represent? It is not clear in text."
>
> Response: q represents the rope configuration, we will elaborate more in the final version.
>
> * “Is rope a 1-dimensional object? Is it not simulated as a 3D object with many degrees of freedom?”
>
> Response: We followed prior works where the rope is described as a 1D object, see [1, 2].
>
> [1] Seita, D., Florence, P., Tompson, J., Coumans, E., Sindhwani, V., Goldberg, K., & Zeng, A. (2021, May). Learning to Rearrange Deformable Cables, Fabrics, and Bags with Goal-Conditioned Transporter Networks. In 2021 IEEE International Conference on Robotics and Automation (ICRA) (pp. 4568-4575). IEEE.,
>
> [2]Sundaresan, P., Grannen, J., Thananjeyan, B., Balakrishna, A., Laskey, M., Stone, K., ... & Goldberg, K. (2020, May). Learning rope manipulation policies using dense object descriptors trained on synthetic depth data. In 2020 IEEE International Conference on Robotics and Automation (ICRA) (pp. 9411-9418). IEEE.
>
> * “What is the inverse model based on? I would suggest adding the network architecture diagram into the text as this would be helpful for a reader trying to follow the text.”
>
> Response: The inverse model appears in the appendix section 8.4 and figure 4. We will reference it inside the paper.
>
> * “What is meant by "The difficulty in manipulating ropes arises from the limited number of actions to change the rope's topological state"?"
>
> Response: Manipulating deformable objects requires very accurate actions, and even small changes in the action can move the rope to a different topological state. In experiment 2, we changed the friction by 5%, and as a result, 18% of the state-action pairs in our dataset led to different topological states than using the original friction.  We thank the reviewer for the comment and will elaborate on it more.
>
> * All text and presentation concerns will be addressed in the final version.
> ‏

---

### Author Response · Authors · 2023-08-09
**General response**

We would like to thank all the reviewers for their helpful comments, and we will address them all in the final version. We respond to the main points here.
Several reviewers asked a few things:
* The space of high-level (topological) states does indeed form a graph (it is large and known in advance), but it is unclear how to manipulate
 the rope (at a low level using notations of configurations and curves) to achieve the high-level motions described in that graph. Therefore,
 our algorithm maintains a tree of configurations (initially rooted in the initial configuration) corresponding to part of that graph that
 previous iterations of the algorithm have discovered. We will clarify this in the final version.
* We will include a more detailed review of knot theory, including topological states and Reidemeister moves in the final version.
* We will provide more details about our data collection schema, the difficulties we overcame, and the limitations of our method.
* All the presentation issues that were raised will be addressed

---

### Decision · Program_Chairs · 2023-08-30

**Decision:**

Accept (Poster)

**Comment:**

The paper presents a hierarchical planning algorithm, TWISTED, for solving the challenging problem of knot tying in rope manipulation. The proposed approach leverages concepts from knot theory and introduces a novel methodology that does not rely on human demonstrations. The algorithm involves two levels of planning: a high-level planner that operates in a discretized space of topological states and a low-level planner that employs an auto-regressive model to transition between these states. The results of the algorithm are evaluated in simulation using the Mujoco platform and compared against various baselines.
Strengths: The paper offers a novel approach to addressing knot tying in rope manipulation through the introduction of a hierarchical planning algorithm. The concept of utilizing topological states to represent the rope's configurations is innovative and demonstrates an effective way to reduce the complexity of the problem. The proposed method exhibits improved performance in simulation, surpassing the performance of various baselines, which underscores the efficacy of the algorithm. The paper demonstrates a systematic approach to evaluating the algorithm by conducting experiments to evaluate the impact of different design choices on its performance.
Weaknesses: While the simulation results are promising, the lack of real-world experimentation is a limitation. It is acknowledged that real-world conditions would present additional challenges, such as perception and control issues, which need to be considered for practical deployment. Some simplifications in terms of perception and control are made, including assuming perfect state estimation and the use of a floating gripper. These assumptions may not hold in real-world scenarios and should be addressed in future work.
The reviewers have raised valid concerns and provided constructive feedback on various aspects of the paper. The authors' responses indicate a thorough understanding of the issues and plans to address them. The revisions made in response to the reviewers' feedback are documented and are aimed at enhancing the clarity, presentation, and understanding of the proposed algorithm. While the absence of real-world experiments and the impact of simplifying assumptions are acknowledged limitations, the novelty and potential of the proposed method make it a valuable contribution to the field of robotic manipulation.